# Effect of Rice Protein Hydrolysates as an Egg Replacement on the Physicochemical Properties of Flaky Egg Rolls

**DOI:** 10.3390/foods9020245

**Published:** 2020-02-24

**Authors:** Yung-Jia Chan, Wen-Chien Lu, Hui-Yao Lin, Zong-Ru Wu, Chen-Wei Liou, Po-Hsien Li

**Affiliations:** 1College of Biotechnology and Bioresources, Da-Yeh University, No. 168 University Rd., Dacun, Chang-Hua 51591, Taiwan; chanyungjia@gmail.com; 2Department of Food and Beverage Management, Chung-Jen Junior College of Nursing, Health Sciences and Management, No. 217, Hung-Mao-Pi, Chia-Yi City 60077, Taiwan; m104046@cjc.edu.tw; 3Fancy International Enterprise Co., LTD., No. 53, Taiping 6th St., Taiping Dist., Taichung City 41165, Taiwan; spark@mfbio.com; 4Department of Medicinal Botanical and Health Applications, Da-Yeh University, No.168, University Rd., Dacun, Changhua 51591, Taiwan; ian44989@gmail.com (Z.-R.W.); weber6229231@gmail.com (C.-W.L.)

**Keywords:** flaky egg roll, papain, physiochemical properties, protein hydrolysates, rice protein

## Abstract

Eggs are linked to some health-related problems, for example, allergy, and religious restrictions, thus the food manufacturer is challenged to find egg replacements and include the physicochemical properties of egg in food. In this study, enzymatic hydrolysis of rice protein was used to produce rice protein hydrolysates (RPHs) for use as an egg replacement in flaky egg rolls. Formulations were control (A), rice protein isolate (B), RPH15 (C), RPH30 (D), and RPH60 (E), respectively. The protein content of formula E increased from 19.69 to 22.18 g/100 g, while carbohydrate and sugar content decreased to 64.12 and 12.26 g/100 g, respectively. Overall amino acid contents significantly increased as compared with formula A. The overall acceptability for sensory evaluation was higher with formula C. The color of the sample was highly affected by the protein-rich ingredients accounting to a Maillard reaction progression and causing a decrease in brightness (L*) and increase in redness (a*). RPHs successfully maintained the functional and physiochemical properties, along with flavor and texture, of flaky egg rolls and could be an egg replacement. These high-value RPHs produced by enzymatic hydrolysis could be beneficial for various applications, particularly food and related industries.

## 1. Introduction

Flaky egg roll is the common name for Phoenix egg rolls in Macau, egg biscuits in Vietnam, and the famous short dough biscuits in Taiwan [1]. Producing flaky egg rolls involves a multiple fold-in method, spreading a thin layer of flour batter on a hot circular metal pan, folding in one-quarter of the cooked batter on the opposite side, next rolling the biscuit around a flat to produce a flaky egg roll. Typically, these products have high sugar and fat; however, using a low protein content of wheat flour would cause reduced dough elasticity and extensibility and varying fracture, taste, and sensory crispy gradients [2].

Foods consumed broadly by the community are active tools for nutrient addition and hence are targeted by a continuously expanding and demanding nutrient-conscious market. Protein-fortified crackers fulfill the requirement for a denser composition of nutrients because these products are made up from the ingredients with low protein contents. Enrichment or fortification of cereal products are conducive to enhance the nutritional properties, upsurge consumption of protein, and make up for insufficiencies of amino acids such as lysine, methionine, and tryptophan [3,4].

Egg is usually considered a “basic ingredient” for several bakery products for proper volume, texture, flavor, and color. Eggs play a vital role in bakery products in binding, leavening, tenderizing, and emulsifying because of unique emulsification, coagulation, foaming, and structure building properties [5]. Nevertheless, the health-related problems of egg such as allergy, and dietary restrictions for example, a lacto-ovo-vegetarian diet or vegan, are challenging to the food manufacturer [6]. Milk, eggs, tree nuts, peanuts, shellfish, sulfites, wheat, soy, and fish are the top allergens established in foods, and food allergy prevalence is increasing in the population worldwide [7]. Hence, food allergens are now an important food safety issue [8].

Rice is considered as one of the significantly vital food crops all over the world [9,10]. In Asian countries, rice is a main and fundamental food source that is contributing as an elementary diet for about 2 billion peoples and is used to produce useful bioactive constituents, such as starch and sugar, resulting in the derivation of rice proteins as byproducts [11]. The functional, nutritional characteristic, and application of rice protein have been extensively researched. Protein digestibility and biological value are higher for rice than wheat, corn, and barley proteins [12]. In addition, rice protein hydrolysates (RPHs) are generally considered to have hypoallergenic [13], anti-oxidative [14], anti-hypertensive [15], anti-cancer [16], and anti-obesity [17] activities. Thus, rice signifies a fascinating source for the formulation and manufacturing of protein enriched food commodities.

Cereal products play a vital role in human daily nutrition. The fortification of bakery commodities with functional ingredients has increased in demand because of the capability to diminish the risk of chronic diseases apart from basic nutritional functions [18]. Incorporation of active ingredients not only contribute to different flavor and colors, but also provides nutrients and health advantages. Rice protein, isolates, or hydrolysates have been used as protein sources ingredients and raw materials to enhance the protein content of many food commodities [18]. Nevertheless, the addition or fortification of functional active ingredients such as rice protein hydrolysate will dramatically affect the physicochemical properties of the dough, and the sensory properties of final goods [18,19]. Different protein extraction methods to extract proteins from rice, such as enzymatic, chemical, and physical methods, have been applied industrially [20,21]. Nevertheless, the application for enzymatic hydrolysis of rice protein was limited. In addition, in terms of the content of amino acids, protein from cereals and legumes are nutritionally reciprocal. Furthermore, protein hydrolysates with more than 88% of protein which are easily digested and absorbed by the human body, and some are equivalent to the protein quality of milk, meats, and eggs, have been used as alternative protein enrichment sources. Hence, it is vital to further investigate and study the enzymatic hydrolysis of rice protein characteristics and applications.

In this study, the rice protein hydrolyzed by papain with variable degrees of hydrolysis was used as an egg replacement to produce nutritionally improved flaky egg rolls. In the meantime, the physicochemical characteristics, nutritional, and sensory properties of flaky egg rolls were examined to establish the possibility to produce a high-quality rice protein hydrolysate enriched flaky egg roll.

## 2. Materials and Methods

### 2.1. Materials

Wheat flour collected of hard red spring wheat and hard red winter wheat in a ratio of 2:3 was provided by Chi-Fa Enterprise (Taichung, Taiwan). The analysis of moisture (A.O.A.C, method 934.01), crude protein (A.O.A.C, method 984.13), and ash content (A.O.A.C. method 923.03) followed the methods of the Association of Official Agricultural Chemists (A.O.A.C.) (1995). Moisture, crude protein, and ash content of wheat flour was 13.52%, 11.74%, and 0.35%, respectively.

Rice protein powder (66.9% of protein, dry basis) was provided by Vedan Enterprise (Taichung, Taiwan). The moisture, fat, and carbohydrate content of rice protein powder was 4.75%, 8.63%, and 7.0%, respectively. All chemicals used in this study were American Chemical Society-certified grade and bought from Sigma-Aldrich Chemical Co. (St. Louis, MO, USA).

### 2.2. Preparation of Rice Protein Isolates (RPIs)

Rice protein powders were dispersed in distilled water in a 1:10 ratio. The pH was then adjusted to pH 10 followed by stirring for 1 h at 50 °C. The slurry was centrifuged at 4000 × *g* for 15 min at room temperature. The pH of the supernatant liquid was adjusted to pH 4 by 1 N hydrochloride acid (HCl). Next, it was stirred for 1 h at 50 °C and centrifuged at 8000 × *g* for 25 min at room temperature. The solid residue (RPI) was collected and dried in a vacuum oven at 50 °C for 12 h [22].

### 2.3. Preparation of RPHs

We used the method of Bandyopadhyay et al. [22], with modification. First, 10 g of RPI was dispersed in 200 mL of distilled water, followed by adjusting the pH to pH 10 by 1 N sodium hydroxide (NaOH), and incubated at 50 °C for 1 h with continuous shaking. After adjustment of the pH to pH 8 and treatment with 0.1% (*w/w*) papain, hydrolysis was carried out at 37 °C for 15, 30, and 60 min with constant shaking. The resulting hydrolysates (RPH15, RPH30, and RPH 60, respectively) were adjusted to pH 8 and the enzyme was quickly inactivated by heating at 95 °C for 5 min. The hydrolysates were then freeze-dried and stored at −20 °C for further analysis.

### 2.4. Flaky Egg Roll Preparation

Flaky egg rolls were produced according to the baker’s recipe, which called for wheat flour (Chi-Fa Enterprise, Taiwan), soybean oil (Taisun Enterprise Co., Ltd., Yuanlin, Taiwan), fine sugar (Taiwan Sugar Corporation, Tainan, Taiwan), corn flour (Sun Food Industrial Company, Taipei Taiwan), lecithin (Taiwan Sugar Corporation, Taiwan), salt (Taiyen Biotech Corporation, Tainan, Taiwan), and water, and was fortified with RPI or different RPHs (Table 1). Formulations were labeled A to E for control (A), RPI (B), RPH15 (C), RPH30 (D), and RPH60 (E). All ingredients were mixed thoroughly into a smooth flowing batter. The circular metal pan was preheated, and then a spoonful of the egg roll batter was spread on the pan. The lid was closed on the pan for baking for 30 s. The flaky egg roll was folded in one-quarter on each side, then the hot biscuit was rolled around a flat to produce a flat-shaped roll [23,24].

### 2.5. Chemical Composition Analysis

The proximate composition analysis of flaky egg rolls, including moisture, crude protein, total fat, carbohydrate, sugar, ash, cholesterol, and dietary fiber, was studied according to the methods of the Association of Official Analytical Chemists (A.O.A.C) [25]. The hot air oven method was used to evaluate the moisture content (A.O.A.C. method 934.01); the method of incineration of samples in a muffle furnace at 550−600 °C was used to analyze the ash content (A.O.A.C. method 923.03); the Kjeldahl method by acid digestion was applied to measure the crude protein content of the sample (A.O.A.C. method 990.03); the Soxhlet’s method (A.O.A.C. method 920.39) was used to quantify the total fat content. Phenol-sulfuric acid assay for total sugar test (A.O.A.C. method 988.12); the enzymatic-gravimetric method for determination of total dietary fiber (A.O.A.C. method 985.29); Liebermann–Burchard method which involved dissolving the sample in chloroform and the absorbance was measured by spectrophotometer at wavelength 640 nm was used for cholesterol analysis. Whereas, the carbohydrates content was calculated by the following Formula (1):Total carbohydrates (g/100g) = 100 − (weight of sample [protein + fat + moisture + ash + fiber] in 100 g of sample).(1)

### 2.6. Characteristics and Texture Analysis

The flaky egg roll diameter (length and width) and thickness were measured by using a Vernier caliper. Hardness and fracturability were analyzed by using the 3-point bend rig (HDP/3 PB) with a 5 kg load cell and the heavy-duty platform of the TAXTPlus texture analyzer (Stable Micro Systems, Godalming, UK). The test speed was set at 3.0 mm s^−1^ and trigger force was automatically set at 50 g. The hardness value, which is defined as maximum force applied, and the fracturability value, which is considered as the distance at the point of break [26], were measured.

### 2.7. Color Analysis

The color of flaky egg rolls was measured with the Hunterlab ColorFlex^®^ EZ colorimeter (Hunterlab, Restin, VA, USA). *L*, a**, and *b** values were determined to analyze changes in quality changes. *L** represented the lightness measurement; *a** characterized the greenness-redness value; while the *b** considered the blueness-yellowness value. The instrument was calibrated with a standard black-and-white ceramic tile before measurement. Color measurements were performed at room temperature in triplicate.

### 2.8. Amino Acid Composition

Flaky egg roll samples were hydrolyzed in 5 mL of 6 M hydrochloride acid (HCl) at 110 °C for 24 h, before cooled to room temperature, hydrolysates were then studied for amino acid content and composition by an amino acid autoanalyzer (Shimadzu LC-30 AD) equipped with C18 column (150 × 4.6 mm, 5 μm; Intact Co., Kyoto, Japan). Then, 20 mM L^−1^ phosphate buffer as mobile phase A, and acetonitrile/methanol/water in a ratio of 45:40:15 *v*/*v*/*v* as mobile phase B, were used to equilibrate the column. The oven temperature was maintained at 40 °C. The instrument was operated following the instructions of the manufacturer and analytical results were generated by using LAB Solutions software (5.54SP 5) [27].

### 2.9. Morphological Observation of Starch Granules

Scanning electron microscopy (SEM) by using the Model ABT-150S system (Topcon Corp., Tokyo, Japan) was used to analyze the microstructure properties of flaky egg rolls with different formulations. Flaky egg rolls coated with gold-palladium (Model JBS-ES 150, Ion sputter coater, Topon Corp., Japan) with 15-kV of accelerating potential was used during morphological analysis. A starch suspension (1% *w*/*w*) was prepared by adding 50% of glycerol. A drip of a starch suspension was allocated on a slide and enclosed with a coverslip. The Olympus BX53 polarized light microscope (PLM) equipped with a CCD camera was used to observe the starch’s granule shape and Maltese cross. The SEM microstructure investigation was performed as described by Indrani et al. [28] with modifications.

### 2.10. Sensory Evaluation

A panel was recruited of students from the Department of Medicinal Botanicals and Health Applications in Da-Yeh University. In total, 30 panelists (15 males, age range 25−32 years) were randomly selected to participate in the sensory evaluation session. The procedure was conducted in a sensory laboratory at room temperature following GB/T 13662−2008 and ISO 4121 criteria to appraise the sensory properties of flaky egg roll in terms of appearance, aroma, taste, texture, and acceptability. The panelists were tutored and trained in the identification and rating scales for the intensity of each attribute before testing. The panelists were explained the requirements and evaluation procedures before the test. Moreover, the correct and suitable terminology used and scoring technique were clarified. The panelists who were insensitive and underperformed were rejected. Five flaky egg rolls from each formulation were cut into approximately 3.0 cm^2^ for every single piece. The randomly selected samples were placed on white, disposable polyform plates and covered with food wrap until testing. Each sample was randomly marked with a three-digit number. The panelists were required to rinse their mouth thoroughly with purified water between samples. The flaky egg rolls were evaluated by quantitative descriptive analysis involving 11-point interval scale with scores from 0 to 10 for each attribute, 0 indicating no value and 10 indicating extremely strong value.

### 2.11. Statistical Analysis

Experimental tests and sample treatments were performed in triplicate. Data are reported as mean (SD). All data were analyzed by single-factor ANOVA to determine the significant effects at *p* < 0.05 among treatments. Additionally, Duncan’s new multiple range test was used to compare treatment means.

## 3. Results

### 3.1. Chemical Composition Analysis

The chemical content of flaky egg rolls fortified with different RPHs formulations is presented in Table 2. Fortification with formula E, RPH60, resulted in the lowest moisture content (2.69%) as compared with control formula A (3.14%). Formula E, with hydrolysis time increased to 60 min reduced mean carbohydrate and dietary fiber content to 64.12 ± 0.17 and 2.95 ± 0.07, respectively. The protein content of formula E was the highest among the five treatments (22.18 ± 0.15). The addition of RPH slightly affected ash content, which was increased with increasing RPH hydrolysis time, and significantly different between formula A with B, C, D, and E. Moreover, the cholesterol content remained steady among formulas. Hence, the fortification with RPH led to reduced moisture, carbohydrate and sugar contents but increased protein content of flaky egg rolls.

### 3.2. Dimensional Quality and Textural Analysis

The characteristics and textural properties of flaky egg rolls fortified with different RPHs are illustrated in Table 3. The length (cm), width (cm), and thickness (cm) of flaky egg rolls substantially decreased from formula A to E. Furthermore, the hardness and fracturability were also greatly decreased from 31.67 ± 1.66 to 19.42 ± 2.95 N and 6.47 ± 0.23 to 5.46 ± 0.36 mm, respectively. The sensory meaning of hardness is defined as the maximal force required to compress a food between the molar teeth, which is an index or a guide of how firm the product is on the initial bite. The quantitative analysis of fracturability involves the initial fractures and gives an indication of the formula’s crumbliness, crispiness, crunchiness, and brittleness. Formula A was the most brittle. Brittle foods usually have a glassy structure with higher sugar content [29].

### 3.3. Amino Acid Composition Analysis

The amino acid composition of flaky egg rolls fortified with different RPHs is summarized in Table 4. Albumin, globulin, glutelin, and prolamin are the four main rice protein amino acids [30]. Rice proteins are primarily recognized in the form of storage organelles called protein bodies. Treating broken rice protein with papain generates protein hydrolysates with diverse protein contents. Moreover, the functionality of food proteins can be altered by enzymatic hydrolysis. The control recipe of flaky egg rolls (formula A) contained a high amount of glutamic acid (16.74 g/100 g protein), followed by proline (6.65 g/100 g protein). However, glutamic acid content was decreased with RPI (15.30 g/100 g of protein), RPH15 (14.07 g/100 g of protein), RPH30 (12.76 g/100 g of protein), and RPH60 (12.16 g/100 g of protein). In contrast, proline and arginine contents ranged from 6.65 to 9.01 and 2.61 to 5.04 g/100 g protein, respectively. The contents of essential amino acids, such as leucine, lysine, phenylalanin, threonine, valine, and tryptophan were increased. Meanwhile, the content of partially essential amino acids, histidine and isoleucine, also raised up. In the case of methionine, the control sample was exhibited by 1.53 g/100 g of protein, while 0.94 g/100 g of protein for RPF60. However, the overall and total amino acid content with RPHs was greatly increased as compared to the control sample. Enzymatic hydrolysis of rice proteins can enhance some functional properties, so hydrolysate-based ingredients are appropriate to be used in a wide range of food applications as compared with inherent protein ingredients [20].

### 3.4. Sensory Evaluation

The mean sensory ratings for flaky egg rolls fortified with different RPHs are showed in Table 5. Sensory attributes such as appearance, aroma, taste, texture, and acceptability of all control and RPI and RPHs fortified flaky egg rolls were evaluated. The scores for taste remained steady across samples. The mean aroma score increased gradually from 6.65 ± 0.32/10 (formula A) to 7.02 ± 0.17/10 (formula E). The addition of RPH greatly improved the aroma. The mean appearance and texture scores ranged from 7.43 ± 0.68/10 to 7.26 ± 0.48/10 and 7.95 ± 0.54/10 to 7.09 ± 0.21/10, respectively. In short, the overall acceptability score was highest in the case of sample RPH15 (7.24 ± 0.09).

### 3.5. Color Analysis

The color changes of flaky egg rolls fortified with different RPHs are exhibited in Figure 1 and flaky egg roll images are demonstrated in Figure 2. Lightness decreased from formula A (122.5) to formula E (112.8). In contrast, a* and b* values sharply increased from formula A to E, from 21.9 to 24.0, and 4.68 to 6.40, respectively. Color development which occurs during the baking process in biscuits is mainly contingent on the Maillard reaction and caramelization of amino acids or proteins and sugars on the surface of the biscuits. Furthermore, during heat treatment of food components, acrylamide could be formed as a result of the Maillard reaction between amino acids and reducing sugars [31]. The occurrence of acrylamide has been linked to a safety concern due to this compound classification as possibly carcinogenic in humans [31]. As well, starch present in wheat flour might also be responsible in part for color development [32]. Our results of RPH-fortified flaky egg rolls agreed with the study confirming that the color was highly affected by polydextrose and protein-rich ingredients, accounting for a Maillard reaction progression and causing a decrease in brightness (L*) and an increase in redness (a*) [33].

### 3.6. Microstructure of Flaky Egg Rolls

Figure 3 revealed the scanning electron micrographs of control (formula A) and flaky egg rolls fortified with RPH15 (formula C). The SEM micrographs for control and formula C egg rolls were relatively similar. Generally, biscuit doughs with high fat content are short doughs and consist of suspended protein, starch–protein corporations, and starch based on lipid emulsion in sugar solution [34]. The RPH compounds being naturally water-soluble are predictable to be in the solution phase of the dough system as well as with some sugar. A previous study suggested that short dough was a bi-continuous system consisting of fat and non-fat phases, in which the non-fat phase formed of sugar solution surrounding the flour (protein and starch). With formula C (RPH15), the matrix of flaky egg rolls contained aggregated proteins, sugars with spread-out fat particles, and ingrained starches. The RPH could be accumulating with proteins or interrelating with sugars existing in the flaky egg rolls during baking. However, a few spaces present in both samples signify a typical chemically leavened egg roll structure with these spaces contributing to a better “mouth-feel” texture of the flaky egg rolls [35].

## 4. Conclusions

RPH hydrolysis by papain confers high protein solubility and emulsifying properties to materials. Papain is a cheap and US Food and Drug Administration approved vegetable-derived enzyme. RPH additions successfully maintained the functional and physiochemical properties of flaky egg rolls, along with the flavor and texture. High addition levels of RPHs produced by enzymatic hydrolysis improved some physical and functional properties of flaky egg rolls, so these hydrolyzation compounds are suitable for application in a wide range of foods, as compared with the inherent protein ingredients. Nonetheless, due to food safety being an important target of this study, further qualitative and quantitative analyses towards the possible formation of acrylamide are needed.

## Figures and Tables

**Figure 1 foods-09-00245-f001:**
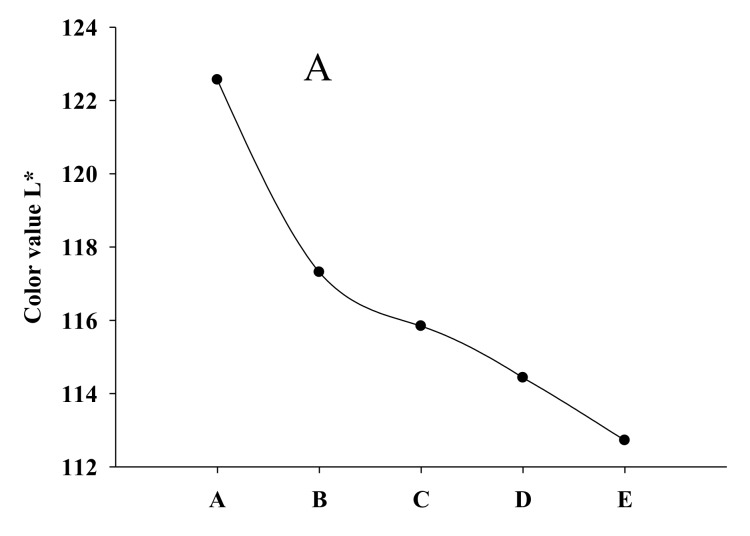
Change in color values for (**A**) *L**, (**B**) *a**, (**C**) *b** of flaky egg rolls fortified with rice protein hydrolysates (RPHs) at different levels. *L** measure lightness, *a** greenness-redness, and *b** blueness-yellowness.

**Figure 2 foods-09-00245-f002:**
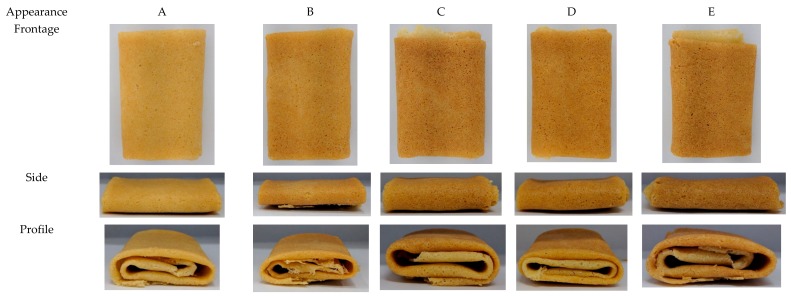
The appearance of flaky egg rolls fortified with RPH at different levels.

**Figure 3 foods-09-00245-f003:**
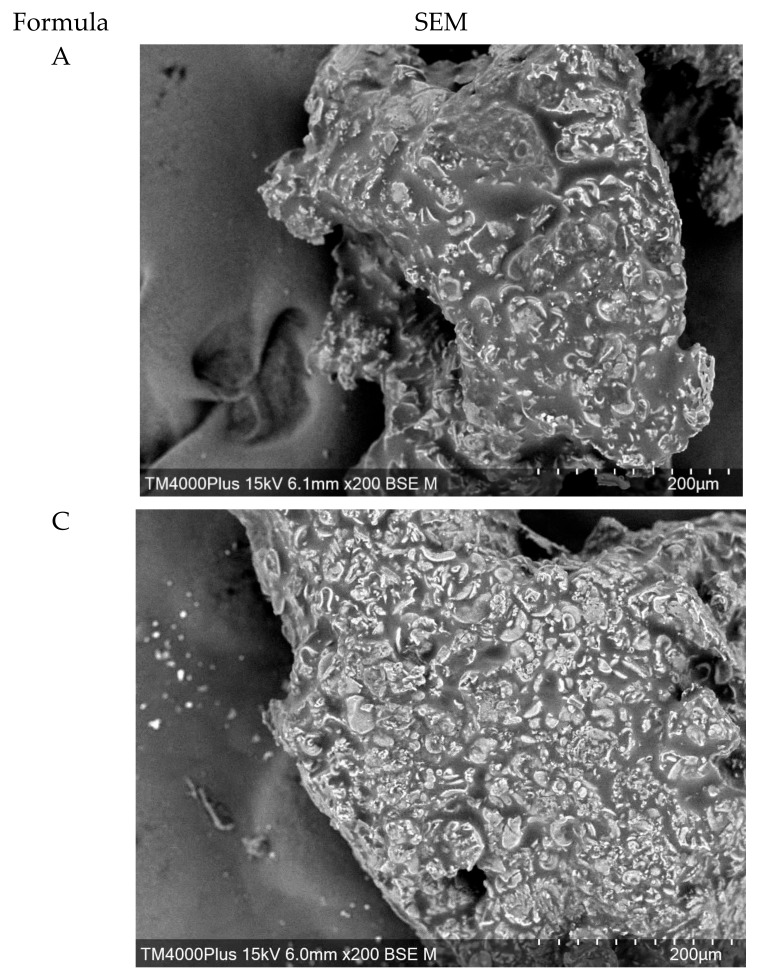
SEM micrographs of flaky egg rolls fortified with RPHs at different levels.

**Table 1 foods-09-00245-t001:** Formula of flaky egg rolls fortified with rice protein hydrolysates (RPHs) at different levels represented in baker’s percentage.

Ingredients	A ^2^	B	C	D	E
Wheat flour	100	100	100	100	100
Oil	85.3	85.3	85.3	85.3	85.3
Sugar	40.9	40.9	40.9	40.9	40.9
Egg	20.5	0	0	0	0
RPI ^1^	0	20.5	-	-	-
RPH15	-	-	20.5	-	-
RPH30	-	-	-	20.5	-
RPH60	-	-	-	-	20.5
Corn flour	5.12	5.12	5.12	5.12	5.12
Lecithin	0.34	0.34	0.34	0.34	0.34
Salt	0.34	0.34	0.34	0.34	0.34
Water	88.7	88.7	88.7	88.7	88.7
Baker’s percentage (%)	341.2	341.2	341.2	341.2	341.2

^1^ RPI: rice protein isolates. ^2^ Formulations were labeled A to E for control (A), RPI (B), and RPI hydrolyzed by papain at 37 °C for 15 (C), 30 (D), and 60 min (E) (RPH15, 30, and 60, respectively).

**Table 2 foods-09-00245-t002:** Chemical composition of flaky egg rolls fortified with different additions of rice protein hydrolysates (RPHs).

Basic Compositions	A ^1^	B	C	D	E
Moisture (g/100 g)	3.14 ^a2^ ± 0.09 ^3^	3.01 ^b^ ± 0.03	2.88 ^c^ ± 0.04	2.76 ^c^ ± 0.05	2.69 ^d^ ± 0.02
Protein (g/100 g)	19.69 ^c^ ± 0.12	20.17 ^c^ ± 0.17	21.22 ^b^ ± 0.24	21.69 ^b^ ± 0.23	22.18 ^a^ ± 0.15
Total fat (g/100 g)	3.15 ^a^ ± 0.04	3.11 ^a^ ± 0.07	3.14 ^a^ ± 0.05	3.09 ^a^ ± 0.06	3.16 ^a^ ± 0.04
Carbohydrate (g/100 g)	66.10 ^a^ ± 0.37	65.49 ^a^ ± 0.58	65.17 ^a^ ± 0.45	64.91 ^b^ ± 0.23	64.12 ^b^ ± 0.17
Sugar (g/100 g)	12.55 ^a^ ± 0.12	12.43 ^a^ ± 0.14	12.37 ^a,b^ ± 0.17	12.31 ^b^ ± 0.15	12.26 ^b^ ± 0.23
Ash (g/100g)	1.62 ^b^ ± 0.03	1.68 ^a^ ± 0.05	1.71 ^a^ ± 0.01	1.72 ^a^ ± 0.03	1.72 ^a^ ± 0.04
Cholesterol (mg/100g)	0.74 ^a^ ± 0.01	0.76 ^a^ ± 0.02	0.77 ^a^ ± 0.02	0.80 ^a^ ± 0.02	0.78 ^a^ ± 0.03
Dietary fiber (mg/100g)	3.17 ^a^ ± 0.03	3.12 ^b^ ± 0.05	3.10 ^b^ ± 0.06	2.98 ^c^ ± 0.04	2.95 ^c^ ± 0.07

^1^Formulations were labeled A to E for control (A), RPI (B), and RPI hydrolyzed by papain at 37 °C for 15 (C), 30 (D), and 60 min (E) (RPH15, 30, and 60, respectively). ^2^ Different letters represent significant difference (*p* < 0.05) amongst the means, as determined by the Duncan’s multiple range test. ^3^ Data are mean ± SD.

**Table 3 foods-09-00245-t003:** Characteristics and textural properties of flaky egg rolls fortified with RPHs at different levels.

Properties	A ^1^	B	C	D	E
Dimensional characteristics					
Length (cm)	6.67 ^a,2^ ± 0.28 ^3^	6.17 ^b^ ± 0.12	6.03 ^b^ ± 0.17	5.97 ^b^ ± 0.11	5.88 ^b^ ± 0.17
Width (cm)	3.94 ^a^ ± 0.25	3.82 ^a^ ± 0.22	3.63 ^b^ ± 0.16	3.58 ^b^ ± 0.06	3.56 ^b^ ± 0.21
Thickness (cm)	1.42 ^a^ ± 0.14	1.36 ^a^ ± 0.15	1.30 ^b^ ± 0.05	1.27 ^b^ ± 0.18	1.25 ^b^ ± 0.13
Textural properties					
Hardness (N)	31.67 ^a^ ± 1.66	30.64 ^a^ ± 1.60	25.03 ^b^ ± 0.58	21.93 ^c^ ± 1.99	19.42 ^c^ ± 2.95
Fracturability (mm)	6.47 ^a^ ± 0.23	6.32 ^a^ ± 0.28	6.03 ^a^ ± 0.18	5.56 ^b^ ± 0.41	5.46 ^b^ ± 0.36

^1^Formulations were labeled A to E for control (A), RPI (B), and RPI hydrolyzed by papain at 37 °C for 15 (C), 30 (D), and 60 (E) min (RPH15, 30, and 60, respectively). ^2^ Different letters represent significant difference (*p* < 0.05) amongst the means, as determined by the Duncan’s multiple range test. ^3^ Data are mean ± SD.

**Table 4 foods-09-00245-t004:** Amino acid composition of flaky egg rolls fortified with RPHs at different levels.

Amino Acid (g/100 g Protein)	A ^1^	B	C	D	E
Essential Amino Acids					
Leucine	4.47	4.59	4.65	4.60	4.74
Lysine	1.90	1.92	1.93	1.89	1.94
Phenylalanin	3.26	3.44	3.57	3.59	3.74
Threonine	3.08	3.12	3.14	3.08	3.15
Valine	2.97	3.26	3.48	3.58	3.79
Methionine	1.53	1.67	1.77	1.81	0.94
Tryptophan	0.62	0.69	0.78	0.77	0.82
Arginine	2.61	3.44	4.09	4.51	5.04
Non-essential Amino Acids					
Tyrosine	2.64	2.86	3.03	3.10	3.27
Cycteine	1.08	0.98	0.90	0.81	0.77
Aspartic acid	3.62	4.78	5.70	6.31	7.06
Serine	3.25	3.28	3.28	3.20	3.27
Glutamic acid	16.74	15.30	14.07	12.76	12.16
Proline	6.65	7.47	8.10	8.42	9.01
Glycine	2.52	2.98	3.33	3.54	3.85
Alanine	2.20	2.57	2.86	3.02	3.28
Partially Essential Amino Acids					
Histidine	1.21	1.36	1.47	1.53	1.63
Isoleucine	2.89	2.99	3.04	3.02	3.12
TAA ^2^	63.24	66.7	69.19	69.54	71.58

^1^Formulations were labeled A to E for control (A), RPI (B), and RPI hydrolyzed by papain at 37 °C for 15 (C), 30 (D), and 60 (E) min (RPH15, 30, and 60, respectively). ^2^ TAA: total amino acids.

**Table 5 foods-09-00245-t005:** Sensory evaluation of flaky egg rolls fortified with RPHs at different levels.

Sensory Properties	A ^1^	B	C	D	E
Appearance	7.43 ^a,2^ ± 0.68 ^3^	7.33 ^a^ ± 0.21	7.27 ^a^ ± 0.35	7.29 ^a^ ± 0.52	7.26 ^a^ ± 0.48
Aroma	6.65 ^b^ ± 0.32	6.71 ^b^ ± 0.24	6.82 ^a^ ± 0.22	6.91 ^a^ ± 0.17	7.02 ^a^ ± 0.17
Taste	7.42 ^a^ ± 0.41	7.41 ^a^ ± 0.11	7.36 ^b^ ± 0.15	7.39 ^a^ ± 0.24	7.40 ^a^ ± 0.07
Texture	7.95 ^a^ ± 0.54	7.17 ^b^ ± 0.25	7.19 ^b^ ± 0.31	7.12 ^b^ ± 0.33	7.09 ^b^ ± 0.21
Acceptability	7.13 ^a^ ± 0.76	7.01 ^a^ ± 0.19	7.24 ^a^ ± 0.09	6.95 ^a^ ± 0.47	6.77 ^b^ ± 0.56

^1^Formulations were labeled A to E for control, RPI, and RPI hydrolyzed by papain at 37 °C for 15, 30, and 60 min (RPH15, 30, and 60, respectively). ^2^ Different letters represent significant difference (*p* < 0.05). ^3^ Data are mean ± SD.

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
