# Peer review of "Effect of Rice Protein Hydrolysates as an Egg Replacement on the Physicochemical Properties of Flaky Egg Rolls"

_foods, 2020, doi:10.3390/foods9020245_

Round 1

Reviewer 1 Report

The manuscript describes an enzymatic hydrolysis of rice protein to be used as an egg replacement in flaky egg rolls. Authors compare protein carbohydrate and sugar contents regard to egg alternative. They also carry out a sensory evaluation concluding that it could be beneficial for various applications in food and related industries. 

Nevertheless, my main concern is that the study is based on the evaluation of a specific food (flaky egg rolls). The study is simply an application of methodologies for evaluating basic information about fundamental nutrients, texture, color, aroma, taste, etc.

Therefore, I recommend its publication but after major revision, where authors includes another food stuff and additional nutritional information.

Author Response

Point 1:

The manuscript describes an enzymatic hydrolysis of rice protein to be used as an egg replacement in flaky egg rolls. Authors compare protein carbohydrate and sugar contents regard to egg alternative. They also carry out a sensory evaluation concluding that it could be beneficial for various applications in food and related industries.

Nevertheless, my main concern is that the study is based on the evaluation of a specific food (flaky egg rolls). The study is simply an application of methodologies for evaluating basic information about fundamental nutrients, texture, colour, aroma, taste, etc.

Therefore, I recommend its publication but after major revision, where authors include another food stuff and additional nutritional information.

Response 1:

Thanks for the positive comments. The replacement of eggs is a trend in future food processing. Moreover, the egg replacement ingredient’s market is expected to gain a critical share of the global food and beverage industry in the future tendency. Increasing trend of vegetarianism and changing diet habits across the globe is a major factor that will help the egg replacement ingredient’s market gain an extensive consumer base.

Previous researchers investigated the possibility of various ingredients as egg substitutes in cereal products. Different types of proteins have been tested as substitutes for eggs in cereal products (Lin, Tay, Yang, Yang, & Li, 2017; Arozarena et al., 2001; Jyotsna, Sai Manohar, Indrani, & Venkateswara Rao, 2007; Rahmati & Mazaheri Tehrani, 2014). However, their success has been limited, suggesting that the substitute proteins have different functions compared with eggs and additional components are needed to make equivalent quality cereal products using eggless recipes.

Improvements in flavour and texture of egg replacement ingredient will enable the market to expand its consumer base. Nevertheless, the production type of egg replacement ingredients, food commodities to be replace or substitute, even the morphological and physicochemical characteristics, and nutritional value of the food after replaced, need to be further discovered. Hence, in this study, we first using the flaky egg rolls (a very general and prevalent cookie in Asia) as the research model to study the rice protein hydrolysates by papain with variable degrees of hydrolysis was used as egg replacement to produce nutritionally improved flaky egg rolls. In the meantime, the physicochemical characteristics, nutritional, and sensory properties of flaky egg rolls was examined to establish the possibility to produce a high-quality rice protein hydrolysates enrichment flaky egg roll.

In addition, we currently plan to study the egg replacement ingredients, rice protein hydrolysate, on other foods, such as noodles, bread or cake products.

Reviewer 2 Report

The topic is interesting, the readability is excellent and the experimental section is complete. The results are encouraging and well reported, by using a correct combination of figures/tables.

The authors should improve the following points:

Please spell-check the MS, some keying errors are present.

Line 51: the point related to blood cholesterol and eggs consumption is controversial. Many authors reported no tight link between eggs consumption and cholesterol issues. The authors should develop this point more accurately.

Line 55: From a food safety point of view, sulphites are important compounds included in allergens list. The authors should add these compounds and references.

Materials and methods: please specify the suppliers of reagents. Please specify how the pH were adjusted (what reagent) and all centrifugation temperatures. Moreover, if available, please add the uncertainty value for each temperature indicated in the Paper. Line 94: …pH 4 by addition of 1N….

(the same at line 99).

Please correct the indications at lines 98 and 149, i.e. We used the method described by Bandyopadhyay et al [20], with modification.

Line 104: …for further analysis.

Line 125: Please complete the list, adding sugar, cholesterol and fiber.

Line 130: mm s-1.

Line 143: Please add more info, i.e. about calibration and sensitivity.

Please check line 148.

Line 156: please specify how the panelists were trained.

Line 174: Please be more specific when talking about shelf-life prolongation.

Please deleted data at line 178, they are not necessary.

Line 179: statistically significant? Please modify as more appropriate.

Please check the font at line 193.

The paragraph 3.3 is very important. Please improve it. Please delete unnecessary data at line 208. Please modify at line 207:….and RPH60 up to 12.16. The part related to essential AAs needs improvement. The increase of some AAs seem not statistically significant, but there are no statistical indications in table 4. The case of methionine is emblematic since the control amount was 1.53 and RPF60 was 0.94. Moreover, many authors listed isoleucine and histidine among “partially essential” AAs. Please add references. Please add a new row “Total” at the end of table 4.

At page 8, the authors made reference to Maillard reaction. Since the improvement of food safety is an important target of this study, considerations/comments about the possible formation of acrylamide are needed. Please remove underlining at line 235 and 237.

The discussion and conclusion need improvement. The authors should clearly identify the formulation which gave the best parameters/acceptability and add comments about all health implications. Moreover, the significance of this study could be enhanced if the authors would try to hypothesize other possible applications of this replacement. These comments could lay the foundations for future studies about this topic.

Please correct in tables footnote: mean ± SD

Author Response

Point 1: Please spell-check the MS, some keying errors are present.

Response 1: Thanks for the comment and positive evaluation. Some grammatical errors, verb tense, singular/plural, and improper words had been modified and revised. Moreover, this paper has been edited by a native English speaker prior to submission.

Point 2: Line 51: the point related to blood cholesterol and eggs consumption is controversial. Many authors reported no tight link between eggs consumption and cholesterol issues. The authors should develop this point more accurately.

Response 2: We agreed with the reviewer’s comment. Eggs contain high amounts of cholesterol, a high dietary intake of which has been linked to cardiovascular diseases. Although the cholesterol limit has been removed from the 2015–2020 Dietary Guidelines for Americans, dietary limitations of cholesterol are still recommended to elderly people or people with previous incidents of heart disease (Aljohi, Dopler-Nelson, Gifuentes, & Wilson, 2019; Kishimoto et al., 2017). Hence, the sentence in line 51 has been re-written and redraft. All changes are marked in yellow background in the revised manuscript.

Point 3: Line 55: From a food safety point of view, sulphites are important compounds included in allergens list. The authors should add these compounds and references.

Response 3: Thank you for the comment. The sentence has been re-written and redraft. All changes are marked in yellow background in the revised manuscript.

Point 4: Materials and methods: please specify the suppliers of reagents. Please specify how the pH were adjusted (what reagent) and all centrifugation temperatures. Moreover, if available, please add the uncertainty value for each temperature indicated in the Paper. Line 94: …pH 4 by addition of 1N…. (the same at line 99).

Response 4: The reference and sentence had been re-written and redraft. We had specified the pH adjustment and centrifugation temperature. All changes are marked in yellow background in the revised manuscript.

Point 5: Please correct the indications at lines 98 and 149, i.e. We used the method described by Bandyopadhyay et al [20], with modification.

Response 5: The reference and sentence had been re-written and redraft at Section 2.3 and Section 2.9. All changes are marked in yellow background in the revised manuscript.

Point 6: Line 104: …for further analysis.

Response 6: Thank you for the comment and positive evaluation. The sentence has been re-written and redraft from “…for further analysis used” to “…for further analysis” in Section 2.3. All changes are marked in yellow background in the revised manuscript.

Point 7: Line 125: Please complete the list, adding sugar, cholesterol and fibre.

Response 7: Thank you for the comment and positive evaluation. We had completed the method used for sugar, cholesterol and fibre analysis in Section 2.5. The sentence has been re-written and redraft. All changes are marked in yellow background in the revised manuscript.

Point 8: Line 130: mm s-1.

Response 8: Thank you for the comment. The sentence has been re-written and redraft. All changes are marked in yellow background in the revised manuscript.

Point 9: Line 143: Please add more info, i.e. about calibration and sensitivity.

Response 9: Thanks for the positive evaluation. The amino acid composition analytical method has been re-written and redraft in more detail in the Section 2.8. All changes are marked in yellow background in the revised manuscript.

Point 10: Please check line 148.

Response 10: Thank you for the comment and positive evaluation. Section 2.9 had been re-written and redraft with more detail and specified analytical method. All changes are marked in yellow background in the revised manuscript.

Point 11: Line 156: please specify how the panellists were trained.

Response 11: Thank you for the comment. The specification of panellists training procedure has been re-written and redraft for more detail in Section 2.10. All changes are marked in yellow background in the revised manuscript.

Point 12: Line 174: Please be more specific when talking about shelf-life prolongation.

Response 12: Thank you very much for the positive comment. After referred to the results and literature reviews, our study does not have enough evidence to prove that the connection between the moisture content and the shelf-life stability. Therefore, the sentence had been re-written and redraft. All changes are marked in yellow background in the revised manuscript.

Point 13: Please deleted data at line 178, they are not necessary.

Response 13: Thanks for the positive comment. The sentence has been re-written and redraft. All changes are marked in yellow background in the revised manuscript.

Point 14: Line 179: statistically significant? Please modify as more appropriate.

Response 14: Thanks for the comment. The sentence has been re-written and redraft. All changes are marked in yellow background in the revised manuscript.

Point 15: Please check the font at line 193.

Response 15: The font at Section 3.2 had been modified. All changes are marked in yellow background in the revised manuscript.

Point 16: The paragraph 3.3 is very important. Please improve it. Please delete unnecessary data at line 208. Please modify at line 207:….and RPH60 up to 12.16. The part related to essential AAs needs improvement. The increase of some AAs seem not statistically significant, but there are no statistical indications in table 4. The case of methionine is emblematic since the control amount was 1.53 and RPF60 was 0.94. Moreover, many authors listed isoleucine and histidine among “partially essential” AAs. Please add references. Please add a new row “Total” at the end of table 4.

Response 16: Thank you very much for the comment. The paragraph 3.3 has been re-written and redraft with the appropriate discussion based on the observed results in the study. All changes are marked in yellow background in the revised manuscript.

Point 17:  At page 8, the authors referred to Maillard reaction. Since the improvement of food safety is an important target of this study, considerations/ comments about the possible formation of acrylamide are needed. Please remove underlining at line 235 and 237.

Response 17: Thank you for the comment. Section 3.5 has been re-written and redraft with the appropriate discussion based on the observed results in the study. All changes are marked in yellow background in the revised manuscript.

Point 18: The discussion and conclusion need improvement. The authors should clearly identify the formulation which gave the best parameters/acceptability and add comments about all health implications. Moreover, the significance of this study could be enhanced if the authors would try to hypothesize other possible applications of this replacement. These comments could lay the foundations for future studies about this topic.

Response 18: Thank you for the comment and positive evaluation. The discussion and conclusion have been re-written and redraft with the appropriate discussion based on the observed results in the study. Some grammatical errors, verb tense, singular/plural, and improper words had been modified and revised. Meanwhile, Table 4 had also been redrafted. All changes are marked in yellow background in the revised manuscript.

Point 19: Please correct in tables footnote: mean ± SD

Response 19: Thank you very much for the comment. The tables footnote has been re-written and redraft. All changes are marked in yellow background in the revised manuscript.

Reviewer 3 Report

Title: change "egg replacement" with "egg partial replacement".

The entire manuscript is heavily influenced by a very low level of English. Many sentences are unclear and their content cannot be evaluated. The authors must have the manuscript reviewed by an English native speaker.

Line 34. After "Taiwan" add a reference.

Line 48. Change "for bakery" to "for several bakery".

Line 52. Clarify the religious restrictions and add a ref. or delete.

Line 70. "These" ingredients. It is unclear: which ingredients? Be more specific and clear (mention the specific ingredients you are referring to). Do you mean rice protein hydorlizate?

Line 72. Ref. 17 seems not appropriate for this sentence. Check and substitute with another one.

Lines 72-77. The use of legumes as protein-rich ingredients is out of the scope of this paper, therefore I suggest to delete all these sentences and refs. therein. Instead, the state of the art on the use of rice protein hydrolizates should be reviewed and reported here.

Line 78. The innovativity of this study has to be clearly stated here, compared to the state of the art. 

Table 1. In the tables and text I suggest to change "A" with "control".

For all the ingredients specify the type (i.e. botanic type of oil, type of corn flour, if refined or not, ...) and the producing company, as well as the main compositional data as reported on the label.

Line 121. Specifiy the method number.

Line 162. Change "for each attribute" to "for the intensity of each attirbute".

Line 176 and 179. Change "composition" to "content".

Table 2. Gve the acronyms of RPH and RPI in the footnote.

Table 3. Change "Characteristic properties" to "Dimensional characteristics".

Author Response

Point 1: The entire manuscript is heavily influenced by a very low level of English. Many sentences are unclear, and their content cannot be evaluated. The authors must have the manuscript reviewed by an English native speaker.

Response 1: We appreciate the positive feedback from the reviewer for a careful and thorough reading of this manuscript and for the thoughtful comments and constructive suggestions, which help to improve the quality of this manuscript. Some grammatical errors, verb tense, singular/plural, and improper words had been modified and revised.

Point 2: Line 34. After "Taiwan" add a reference.

Response 2: Thank you very much for the comment. The reference had been added in line 35. All changes are marked in yellow background in the revised manuscript.

Point 3: Line 48. Change "for bakery" to "for several bakery".

Response 3: Thank you for the comment and positive evaluation. The sentence has been re-written and redraft from "…for bakery" to "…for several bakery". All changes are marked in yellow background in the revised manuscript.

Point 4: Line 52. Clarify the religious restrictions and add a ref. or delete.

Response 4: The sentence has been re-written and redraft. All changes are marked in yellow background in the revised manuscript.

Point 5: Line 70. "These" ingredients. It is unclear: which ingredients? Be more specific and clearer (mention the specific ingredients you are referring to). Do you mean rice protein hydrolysate?

Response 5: The sentence was referred to the rice protein hydrolysate. Due to the unclear sentence, the sentence has been re-written and redraft. All changes are marked in yellow background in the revised manuscript.

Point 6: Line 72. Ref. 17 seems not appropriate for this sentence. Check and substitute with another one.

Response 6: Thank you for the comment. The references and sentence had been re-written and redraft with more suitable literature reviews. All changes are marked in yellow background in the revised manuscript.

Point 7: Lines 72-77. The use of legumes as protein-rich ingredients is out of the scope of this paper, therefore I suggest deleting all these sentences and refs. therein. Instead, the state of the art on the use of rice protein hydrolizates should be reviewed and reported here.

Response 7: Thanks for the comment. The references and sentences have been re-written and redraft. Some grammatical errors, verb tense, singular/plural, and improper words had been modified and revised. All changes are marked in yellow background in the revised manuscript.

Point 8: Line 78. The innovatively of this study must be clearly stated here, compared to the state of the art.

Response 8: Thank you for the comment and positive evaluation. The objective of the study, in Section 1, had been re-written and redraft. All changes are marked in yellow background in the revised manuscript.

Point 9: Table 1. In the tables and text, I suggest to change "A" with "control".

Response 9: Thanks for the comment. A footnote of Table 1 highlights and revealed the formulation of flaky egg roll of formula A as control was stated at Table 1. All changes are marked in yellow background in the revised manuscript.

Point 10: For all the ingredients specify the type (i.e. botanic type of oil, type of corn flour, if refined or not, ...) and the producing company, as well as the main compositional data as reported on the label.

Response 10: The ingredient specification, such as the producing company details, and sentence has been re-written and redraft. All changes are marked in yellow background in the revised manuscript.

Point 11: Line 121. Specify the method number.

Response 11: Thank you for the comment. In Section 2.5, the AOAC method number had been added in. The sentence has been re-written and redraft. All changes are marked in yellow background in the revised manuscript.

Point 12: Line 162. Change "for each attribute" to "for the intensity of each attribute".

Response 12: Thank you for the comment. The sentence has been re-written and redraft from "…for each attribute" to "…for the intensity of each attribute". All changes are marked in yellow background in the revised manuscript.

Point 13: Line 176 and 179. Change "composition" to "content".

Response 13: Thank you for the comment and positive evaluation. The sentence has been re-written and redraft in the Section 3.1. All changes are marked in yellow background in the revised manuscript.

Point 14: Table 2. Give the acronyms of RPH and RPI in the footnote.

Response 14: Thank you for the comment and positive evaluation. The footnote of Table 2 has been re-written and redraft. All changes are marked in yellow background in the revised manuscript.

Point 15: Table 3. Change "Characteristic properties" to "Dimensional characteristics".

Response 15: The sentence and words had been re-written and redraft from "Characteristic properties" to "Dimensional characteristics". All changes are marked in yellow background in the revised manuscript.

Round 2

Reviewer 2 Report

The authors improved their MS, following the suggestions/comments.

Minor revision required.

Line 140: Please add the reference for this method.

Line 142: Other info is needed. Please specify the type and supplier of GC system and add info about the calibration procedures adopted for this determination.

Line 165: mM L-1

Line 166: …was used as mobile phase B…..

Line 246: …Interestingly, the case of methionine………

Line 278: Please add a brief comment about the safety aspects linked to acrylamide formation.

Please correct the spacing in tables footnote: mean ± SD.

Author Response

We appreciate the reviewer comments very much and have revised the manuscript accordingly. The specific changes we have made in the revised manuscript are, as follows:

Point 1: The authors improved their MS, following the suggestions/comments.

Response 1: We appreciate the positive feedback from the reviewer for a careful and thorough reading of this manuscript and for the thoughtful comments and constructive suggestions, which help to improve the quality of this manuscript. Some grammatical errors, verb tense, singular/plural, and improper words had been modified and revised.

Point 2: Minor revision required.

Response 2: The MS had been modified and revised. All changes are marked in yellow background in the revised manuscript.

Point 3: Line 140: Please add the reference for this method.

Response 3: Thank you for the comment. In Section 2.5, the reference of the method had been added in. The sentence has been re-written and redraft. All changes are marked in yellow background in the revised manuscript.

Point 4: Line 142: Other info is needed. Please specify the type and supplier of GC system and add info about the calibration procedures adopted for this determination.

Response 4: Thanks for the positive evaluation. The analytical method has been re-written and redraft in more detail in the Section 2.5. All changes are marked in yellow background in the revised manuscript.

Point 5: Line 165: mM L-1

Response 5: Thank you for the comment. The sentence has been re-written and redraft. All changes are marked in yellow background in the revised manuscript.

Point 6: Line 166: …was used as mobile phase B…..

Response 6: Thank you for the comment and positive evaluation. The sentence has been re-written and redraft from “…as mobile phase B was used to…” to “…was used as mobile phase B…” in Line 166. All changes are marked in yellow background in the revised manuscript.

Point 7: Line 246: …Interestingly, the case of methionine………

Response 7: Thank you for the comment and positive evaluation. The sentence has been re-written and redraft from “…Interestingly, in the case of methionine…” to “…Interestingly, the case of methionine…” in Line 246. All changes are marked in yellow background in the revised manuscript.

Point 8: Line 278: Please add a brief comment about the safety aspects linked to acrylamide formation.

Response 8: Thanks for the positive comment. A sentence about the safety aspects linked to acrylamide formation in Section 3.5 has been re-written and redraft. All changes are marked in yellow background in the revised manuscript.

Point 9: Please correct the spacing in tables footnote: mean ± SD.

Response 9: Thank you very much for the comment. The tables footnote has been re-written and redraft. All changes are marked in yellow background in the revised manuscript.